# Long-term outcomes of antenatal corticosteroids for preterm birth: An overview of systematic reviews

Saima Sultana[1,2]*, Fiona Bruinsma[2], Zeshi Fisher[2], Caroline S. E. Homer[2], Joshua P. Vogel[2]

1 School of Public Health and Preventive Medicine, Monash University, Melbourne, Australia, 2 Women's, Children's and Adolescents' Health Program, Burnet Institute, Melbourne, Australia

* saima.sultana@monash.edu

## Abstract

Antenatal corticosteroids (ACS) is a critical preterm birth intervention that reduces neonatal mortality and short-term morbidities when administered to women at risk of preterm birth <34 weeks' gestation. While numerous systematic reviews have evaluated the effects of ACS on child health outcomes, this body of evidence has not been comprehensively assessed. This overview of systematic reviews thus aimed to comprehensively summarize the findings on the effects of ACS for preterm birth on long-term outcomes in children. We searched six electronic databases, with no date and language restrictions. Systematic reviews of randomized, non-randomized trials or observational studies that evaluated childhood health outcomes (assessed at age 12 months or older) for any ACS exposure compared to no ACS exposure, including comparisons between different types and regimens of ACS were eligible. We used the AMSTAR-2 tool to assess the quality of the included reviews. We included 19 systematic reviews. We identified a wide range of outcomes across the included reviews and categorized them as neurodevelopmental, psychological, physical growth, respiratory, cardiovascular, and survival/mortality outcomes. The reported outcomes were substantially varied in terms of operational definitions, terminology, timing of assessment, and measurement. No benefits or harms were observed for most of the reported outcomes following ACS exposure, though many outcomes had few participants. Available evidence suggested that a single course of ACS possibly has positive effects on selected neurodevelopmental outcomes. However, ACS exposure might also have adverse effects on psychological and some neurodevelopmental outcome in children born at late preterm or at term. Heterogeneity in outcome measurements and reporting, including overall confidence in results, contributes to uncertainties about long-term effect of ACS and warrants caution while interpreting the findings. Further high-quality research is required to generate robust evidence base on the effects of ACS on long-term child health outcomes.

**Data availability statement:** This is a review of publicly available published data. No new data were created in this study. All data related to this review are presented in the article and supporting information files.

**Funding:** No funding was secured for this study. SS is supported by Monash Graduate Scholarship, JPV is supported by a NHMRC Investigator Grant (GNT1194248) and CH is supported by a NHMRC Fellowship (APP1137745).

**Competing interests:** The authors have declared that no competing interests exist.

## Introduction

Preterm birth is defined as birth before 37 completed weeks of gestation [1]. Each year, an estimated 13.4 million babies are born preterm globally, equating to approximately 10% of all births [2]. Nearly 1 million deaths occur annually due to preterm birth-related complications, which constitutes the leading cause of death globally in children under 5 years [3]. Infants born preterm are more likely to have a wide range of short and long-term morbidities, including respiratory complications, neurodevelopmental disorders, behavioural problems, psychiatric conditions, attention-deficit hyperactivity disorder, and poorer academic performance [4–9]. The risks of both short- and long-term complications are greatest in the earliest gestations.

Antenatal corticosteroids (ACS) remains the cornerstone intervention for improving short-term outcomes of preterm birth [10]. ACS is usually a course of betamethasone or dexamethasone injections that are administered intramuscularly [11]. Further, for women who haven't delivered seven or more days after an initial course of ACS, a repeat course of ACS is also recommended when preterm birth is expected or planned within the next seven days [12–14]. When administered antenatally, ACS can rapidly cross the placenta and expedite maturation of fetal lung and other organ systems. The fetal lung maturational effects are the main mechanism in how ACS promotes improvements in survival and reduces morbidity for preterm babies [15].

The available trial evidence on ACS shows compelling evidence of these short-term health benefits. The 2020 update of the Cochrane review [16] on ACS efficacy reported that ACS administered to women at risk of preterm birth prior to 34 weeks' gestation significantly reduces the risk of neonatal mortality (RR 0.78, 95% CI 0.70-0.87). It also reduces the risk of respiratory distress syndrome (RR 0.71, 95% CI 0.65 to 0.78) and intraventricular hemorrhage (RR 0.58, 95% CI 0.45 to 0.75). The review did not identify any newborn or maternal harms associated with its use. Further evidence from a Cochrane review on repeat doses of ACS suggests that a course of ACS followed by a single repeat course (if required) significantly reduces the risk of respiratory distress syndrome (RR 0.82, 95% CI 0.74 to 0.90) [17]. The use of additional (or multiple) repeat courses does not confer additional benefit. A notable limitation of the evidence base in these reviews is the absence of trial data on the longer-term effects of ACS, such as into childhood or adulthood.

A number of non-randomized studies and systematic reviews have sought to determine the effects of ACS on longer-term outcomes, namely survival, growth, and neurodevelopmental outcomes in children following fetal ACS exposure. Available systematic reviews vary in terms of the interventions considered, outcome measures and conclusions. These somewhat conflicting conclusions pose a challenge for clinicians, women and other stakeholders, as to what the state of knowledge is regarding longer-term benefits or potential harms of ACS administration for preterm birth. Thus, an overview of the reviews is crucial to synthesize the available evidence on the long-term effects of ACS exposure in children. Such an overview would aid researchers, clinicians, and policymakers in decision-making, as well as highlight

knowledge gaps to be investigated in further research activities. Therefore, we aimed to summarize the findings of systematic reviews on long-term outcomes in children associated with ACS for preterm birth.

## Materials and methods

This overview of systematic reviews was carried out following the Cochrane methods for performing an overview of systematic reviews [18]. Our protocol was prospectively registered on PROSPERO (CRD42023475595).

### Eligibility criteria

We included systematic reviews of randomized controlled trials (RCTs), non-randomized trials, and observational studies (with or without meta-analysis) that evaluated long term child health outcomes following ACS exposure compared to no ACS exposure or the use of different ACS types and regimens. We defined 'Long-term outcome' as any outcome occurring/assessed between ages of 12 months to less than 18 years. Eligible interventions were (i) a single course of ACS (dexamethasone or betamethasone) compared to placebo or no treatment, (ii) repeat dose(s) of ACS compared to a single course of ACS, (iii) comparisons between different types of ACS (e.g., dexamethasone versus betamethasone) and (iv) comparisons between different ACS regimens (e.g., dosing, frequency, and timing). We excluded systematic reviews reporting on the use of ACS in animal models, non-systematic reviews, review comments, guidelines, expert committee opinions, statements, and conference abstracts.

### Information sources, search strategy and selection of reviews

In consultation with an experienced academic librarian we developed a comprehensive search strategy and searched Medline, EMBASE, CINAHL, Web of Science, Cochrane library, and Epistemonikos. The initial search was conducted in February 23 & 24, 2023 and it was updated on March 11, 2024. There were no limits on year of publication and language. The full search strategy is available in S1 Appendix. We also screened the reference lists of all included systematic reviews for any additional records. Where there were multiple versions of the same review, we only included the latest version of the review. Using Covidence [19], two reviewers independently screened all titles/abstracts and potentially eligible full texts for inclusion according to the eligibility criteria. Any disagreements were resolved through discussion or consultation with a third reviewer.

### Data extraction and management

Two reviewers (SS, and FB/ZF) independently extracted data using an excel spreadsheet. Any disagreement or uncertainty were resolved by discussion. We extracted data on review characteristics (year of publication, search date, intervention of interest, number of included studies, study design and location of included studies, eligible participants), participant characteristics (gestational age at birth, and timing of outcome assessment), and outcomes reported in the systematic reviews including effect estimates. For reviews including studies on newborns, children and adult populations, we extracted information on child outcomes only. The details of extraction are available in S2 Appendix.

### Methodological quality assessment of included reviews

Two reviewers (SS, and FB/ZF) independently assessed quality of the included reviews using the AMSTAR 2 tool [20]. Any disagreements were resolved by consensus or by consulting a third reviewer. Systematic reviews were assessed across all 16 domains of the AMSTAR 2 tool and assigned an overall quality judgement based on the overall confidence in the review results (i.e., high, moderate, low, critically low).

## Data synthesis

We described the characteristics of the included reviews and synthesized the outcome data narratively. We did not re-analyse data from published reviews. We present the pooled effect estimates, including 95% confidence intervals of the child health outcomes as reported in the included reviews within each comparison. Where available, we also reported the certainty of evidence for each outcome as per the Grading of Recommendations, Assessment, Development and Evaluation (GRADE) [21] analysis as conducted by the authors of the included systematic reviews. We also developed descriptive forest plots for meta-analysed outcomes from included reviews, within each comparison. For those reviews where, meta-analyses were not performed and outcome measures were reported in a narrative form only, we reported review results narratively only.

We also developed a citation matrix to visually demonstrate the overlap among primary studies within included systematic reviews.

## Results

The searches identified 8148 citations. After removal of duplicates, we screened 5923 citations by title and abstract against the pre-specified eligibility criteria. Of these, 165 citations were retained for full-text assessment; 146 were subsequently excluded S3 Appendix, leaving 19 systematic reviews for inclusion (Fig 1).

### Characteristics of included reviews

Characteristics of the 19 included reviews are summarized in Table 1. They were published between 1995 and 2023. Three were Cochrane reviews [16,17,37] and 16 were non-Cochrane reviews [22–36,38], including one scoping review, one systematic review and network meta-analysis, and one systematic review and individual participant data meta-analysis. Twelve reviews performed meta-analyses for child health outcomes [16,17,37,27,22,26,34,38,23,35,24,31]. Eight reviews included only RCTs, seven reviews included only observational studies and three reviews included both RCTs and observational studies. In total, the 19 reviews included follow-up data from 13 trials and 62 observational studies (mainly retrospective or prospective cohort studies) published between 1972 and 2022. These primary studies were conducted across 32 countries, with the majority in high-income countries (69%). Six reviews reported the effects of a single course of ACS (betamethasone or dexamethasone) versus placebo/no treatment [16,26,22,25,23,24], five reviews evaluated the effects of repeat/multiple courses of ACS versus a single course of ACS [17,36,34,33,35], two reviews compared between different type of ACS (i.e., betamethasone and dexamethasone) [37,38], and seven reviews reported effects of any ACS exposure (i.e., single/unspecified/repeat/multiple course) versus placebo/no treatment [27,28,22,29,30,31,32].

AMSTAR 2 assessments were performed for the 19 included reviews. Six (31.6%) were rated as of critically low methodological quality [25,26,29,30,35,36], eight reviews were rated as of low to moderate quality (four each) [22–24,27,28,31–33] and five reviews (31.6%) were rated as having high quality [16,17,34,37,38]. Full AMSTAR-2 assessments are reported in S1 Table.

We revealed a high overlap of primary studies across the reviews. The citation matrix of the primary studies is available in S3 Appendix.

### Childhood health outcomes and effect of interventions

A wide range of childhood health outcomes were reported in the included reviews. We classified the reported outcomes into six broad categories – neurodevelopmental, psychological, physical growth, respiratory, cardiovascular, and mortality/survival outcomes. The outcomes we identified are summarized in S2 Table. Summaries of reported meta-analysed outcomes within each comparison are presented in Figs 2–5 and S3 Table.

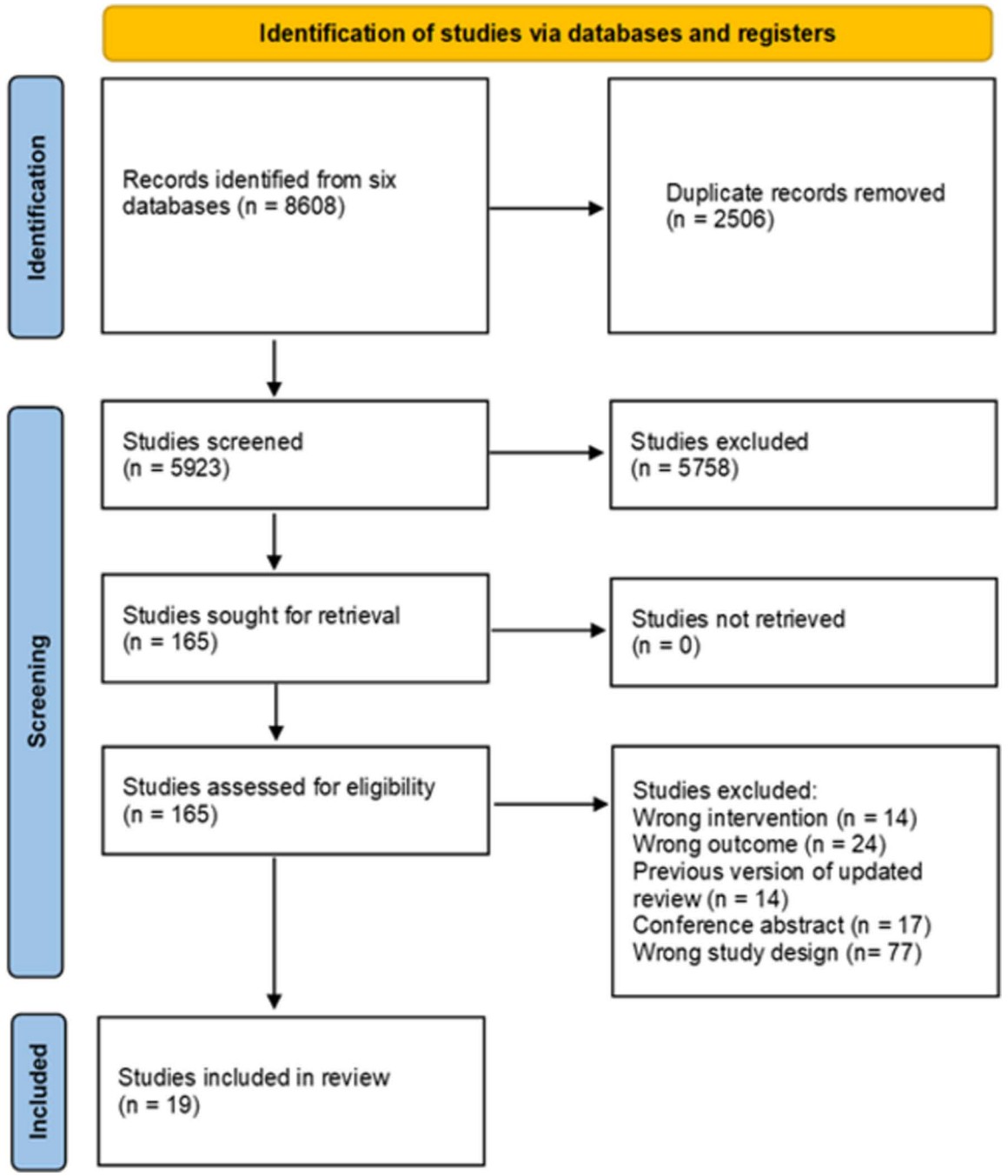

**Fig 1. PRISMA flow diagram.**

**Single course of ACS versus placebo/no treatment.** *Neurodevelopmental outcomes.* In total, 22 neurodevelopmental outcomes were reported across six reviews [16,22–26]. A 2020 Cochrane review [16] reported that compared to placebo, ACS exposure might be associated with a reduction in developmental delay in children (RR 0.51, 95% CI 0.27 to 0.97, three studies, 600 children aged between 2 and 12 years, moderate certainty evidence).

**Table 1. Characteristics of included reviews.**

| Review ID | Type of review | Review objective/s | No. of included studies with child health outcomes | Study design | Country of primary studies | Country income level | Intervention | Comparator |
|---|---|---|---|---|---|---|---|---|
| **Single course of ACS vs. placebo/no treatment** | | | | | | | | |
| McGoldrick 2020 [16] | Systematic review and meta-analysis | To assess the effects of administering a course of corticosteroids to women prior to anticipated preterm birth (before 37 weeks of pregnancy) on fetal and neonatal morbidity and mortality, maternal mortality and morbidity, and on the child in later life | 5 | Randomized controlled trial | Brazil, USA, Finland, New Zealand, Netherlands | HIC, UMIC | Single course of ACS (Betamethasone/ Dexamethasone) | Placebo/ no treatment |
| Ninan 2022 [22] | Systematic review and meta-analysis | To conduct a systematic review and meta-analysis of long-term outcomes associated with preterm exposure to antenatal corticosteroids compared with no exposure in all children as well as children with preterm and full-term birth | 30 | Retrospective cohort, prospective cohort, regression discontinuity | USA, Canada, Singapore, Korea, Spain, China, Japan, Taiwan, Finland, Turkey, France, Sweden, Austria | HIC, UMIC | Single/ unspecified course of ACS (Betamethasone/ Dexamethasone) | No ACS |
| Park 2016 [23] | Systematic review and meta-analysis | To evaluate the effectiveness of antenatal corticosteroids compared with placebo or no treatment in neonates born before 24 weeks of gestation | 3 | Retrospective cohort, prospective cohort | Australia, USA | HIC | Single course of ACS (Betamethasone/ Dexamethasone) | Placebo/ no treatment |
| Sotiriadis 2015 [24] | Systematic review and meta-analysis | To assess the effects on neurodevelopmental outcome of children after administration of a single course of antenatal corticosteroids for threatened preterm labor | 14 | Randomized controlled trial, retrospective cohort, prospective cohort, case control, non-randomized cohort | Australia, USA, China, Netherlands, Finland, New Zealand | HIC | Single course of ACS (Betamethasone/ Dexamethasone) | Placebo/ no treatment |
| Onland 2011 [25] | Systematic Review & Meta-analysis | To determine the short- and medium long-term effects of antenatal corticosteroids on fetal and neonatal mortality and pulmonary and neurodevelopmental sequelae | 1 | Randomized controlled trial | Finland | HIC | Single course of ACS (Betamethasone/ Dexamethasone) | Placebo |
| Crowley 1995 [26] | Systematic review and meta-analysis | To assess the effects of antenatal corticosteroids therapy when given to women before anticipated preterm delivery (elective, or after spontaneous labour) | 3 | Randomized controlled trial | USA, Netherland, New Zealand | HIC | Any acceptable ACS regimen (Betamethasone/ Dexamethasone) | Placebo |

*(Continued)*

**Table 1.** (Continued)

| Review ID | Type of review | Review objective/s | No. of included studies with child health outcomes | Study design | Country of primary studies | Country income level | Intervention | Comparator |
|---|---|---|---|---|---|---|---|---|
| **Any ACS exposure (single/multiple/repeat courses) vs. placebo/no ACS** | | | | | | | | |
| Amiya 2016 [27] | Systematic review and meta-analysis | To assess effects on maternal and child outcomes of antenatal corticosteroids administration among special subgroups of women at risk of imminent preterm birth, including those (1) with pre-gestational and gestational diabetes mellitus, (2) undergoing elective caesarean section in late preterm (34 to < 37 weeks), (3) with chorioamnionitis, and (4) with growth restricted fetuses | 2 | Prospective cohort, case control | Australia, Netherlands | HIC | Single/ repeat course of ACS (Betamethasone/ Dexamethasone) | Placebo/ no treatment |
| Blankenship 2020 [28] | Systematic review and meta-analysis | To estimate the effect of antenatal corticosteroid administration on neonatal mortality and morbidity in preterm small-for-gestational age (SGA) infants through a systematic review and meta-analysis | 3 | Retrospective cohort, case control | Netherlands, Canada, Japan | HIC | Betamethasone | No ACS |
| Sacco 2022 [29] | Systematic review | To systematically review the human clinical literature to determine the effects of ACS on offspring cardiovascular function | 6 | Randomized controlled trial, cohort | New Zealand, Netherlands, Australia, Canada, USA | HIC | Any ACS exposure | Placebo/ no ACS/ population data |
| Sarid 2022 [30] | Scoping review | To synthesize the body of knowledge on the association between ACS exposure for risk of preterm birth and brain development in infants ultimately born late preterm and term | 14 | Cross sectional, retrospective cohort, prospective cohort | Germany, Switzerland, Spain, USA, Finland, Canada | HIC | Single/ multiple courses of ACS (Betamethasone/ Dexamethasone) | No ACS |
| Wang 2022 [31] | Systematic review and meta-analysis | To explore the associations between antenatal corticosteroids exposure and hearing loss in premature infants | 7 | Retrospective cohort, case control | USA, Korea, Japan | HIC | Single/ multiple courses of ACS | no ACS exposure |
| Ninan 2023a [32] | Systematic review and meta-analysis | To systematically review the proportions of infants with early exposure to antenatal corticosteroids but born at term or late preterm, and short term and long-term outcomes. | 4 | Population based cohort | Canada, USA, Finland | HIC | Unspecified courses of ACS | No ACS exposure |

*(Continued)*

| Review ID | Type of review | Review objective/s | No. of included studies with child health outcomes | Study design | Country of primary studies | Country income level | Intervention | Comparator |
|---|---|---|---|---|---|---|---|---|
| **Repeat/multiple courses of ACS vs. single course of ACS** | | | | | | | | |
| Ninan 2023b [33] | Systematic Review | To determine the long-term neurodevelopmental and psychological outcomes following preterm exposure to multiple courses versus a single course of antenatal steroids | 8 | Randomized controlled trial, prospective cohort | USA, Finland, UK, Iran, India, Argentina, Bolivia, Spain, Brazil, Canada, Chile, China, Colombia, Denmark, Germany, Hungary, Israel, Jordan, Netherlands, Peru, Poland, Russian, Switzerland | HIC, UMIC, LMIC | Multiple course of ACS (Betamethasone/ Dexamethasone) | Single course of ACS |
| Walters 2022 [17] | Systematic review and meta-analysis | To assess the effectiveness and safety, using the best available evidence, of a repeat dose(s) of prenatal corticosteroids, given to women who remain at risk of preterm birth seven or more days after an initial course of prenatal corticosteroids with the primary aim of reducing fetal and neonatal mortality and morbidity | 4 | Randomized controlled trial | New Zealand, Australia, USA, Finland, UK, Brazil, Argentina, Bolivia, Canada, Chile, China, Colombia, Denmark, Germany, Hungary, Israel, Spain, Jordan, Netherlands, Peru, Poland, Russia, Switzerland | HIC, UMIC, LMIC | Repeat doses of ACS (Betamethasone/ Dexamethasone) **all included trials used betamethasone | Single course of ACS |
| Crowther 2019 [34] | Systematic review and individual participant data meta-analysis | To assess the effects of repeat prenatal corticosteroid treatment given to women at risk of preterm birth to benefit their infants, both short-term and long-term, and whether the treatment effects differed participant and treatment factors | 6 | Randomized controlled trial | UK, Canada, Finland, Australia, New Zealand, USA | HIC | Repeat doses of ACS (Betamethasone/ Dexamethasone) | Placebo/ no repeat treatment |
| Peltoniemi 2011 [35] | Systematic Review & Meta-analysis | To systematically review the efficacy and safety of repeated antenatal corticosteroid on neonatal morbidity, growth and later development | 4 | Randomized controlled trial | Australia, New Zealand, Finland, USA, Canada | HIC | Repeated ACS (Betamethasone/ Dexamethasone) | Single course of ACS |
| Aghajafari 2001 [36] | Systematic Review & Meta-analysis | To determine the effects of multiple courses of antenatal corticosteroids on perinatal, infant and maternal health outcomes | 1 | Prospective cohort | Australia | HIC | Multiple courses of ACS (Betamethasone/ Dexamethasone) | Single course of ACS |

*(Continued)*

**Table 1.** (Continued)

| Review ID | Type of review | Review objective/s | No. of included studies with child health outcomes | Study design | Country of primary studies | Country income level | Intervention | Comparator |
|---|---|---|---|---|---|---|---|---|
| **ACS vs. ACS** | | | | | | | | |
| Williams 2022 [37] | Systematic review and meta-analysis | To assess the effects on fetal and neonatal morbidity and mortality, on maternal morbidity and mortality, and on the child and adult in later life, of administering different types of corticosteroids (dexamethasone or betamethasone), or different corticosteroid dose regimens, including timing, frequency and mode of administration | 2 | Randomized controlled trial | Australia, New Zealand, France | HIC | Dexamethasone | Betamethasone |
| Ciapponi 2021 [38] | Systematic review & network meta-analysis | To evaluate the comparative clinical effectiveness and safety of dexamethasone vs betamethasone for preterm birth | 3 | Randomized controlled trial | Australia, France | HIC | Dexamethasone | Betamethasone |

HIC = High income country; LMIC = Lower middle-income country; UMIC = Upper middle-income country

A 2022 systematic review of observational studies [22] reported that exposure to a single course of ACS compared to non-exposure was associated with a decrease in neurodevelopmental impairment (aOR 0.69, 95% CI 0.57 to 0.84, two studies, 3948 children aged 18–22 months, low certainty evidence). ACS exposure also showed a reduction in cerebral palsy among preterm born children assessed between 18 months and 12 years, as reported in two systematic reviews [22,24] as well as a reduction in severe disability (RR 0.79, 95% CI 0.73 to 0.85, 5 studies, 6051 children) and an increase in intact survival (i.e., normal neurodevelopment) in children (RR 1.19, 95% CI 1.06 to 1.33, 6 studies, 2644 children) [24]. Another systematic review [23] further reported that ACS exposure might lead to a reduction in the odds of moderate severe cerebral palsy, blindness and deafness (in children who were born before 24 weeks of gestation. There were no clear differences between the single course of ACS and placebo/no ACS group for other neurodevelopmental outcomes reported on the reviews.

*Psychological outcomes.* One review reported no clear difference between the single ACS course and the placebo group for childhood behavioural or learning difficulties after ACS exposure [16].

*Physical growth outcomes.* Two reviews reported childhood outcomes related to physical growth in children [16,22] The reviews found no clear difference in growth outcomes assessed between ages 18 months to 12 years (e.g., height, weight, and head circumference) between the single course ACS group and placebo/no ACS group.

*Respiratory and cardiovascular outcomes.* Four respiratory outcomes were reported in two reviews [22,23] and a one cardiovascular outcome was reported in one review [16]. One review found no difference between the two groups for chronic lung disease at 2 years [23], while a second review reported an increased incidence of respiratory outcomes (asthma, allergic rhinitis) in the ACS group assessed between 2–5 years [22]. There was no clear evidence of a difference in cardiovascular outcome between the single course ACS group and placebo/no ACS group.

*Survival/mortality outcomes.* Two reviews reported outcomes on mortality in children. One review reported that compared to placebo or no treatment, a single course of ACS reduced the odds of child mortality at 18–22 months among those born <24 weeks' gestation [23]. However, the other review found no clear difference between the two groups in childhood death (up to 6 years) [16].

**Any ACS exposure (single/repeat/ multiple/unspecified courses) versus placebo/no ACS.** *Neurodevelopmental outcomes.* Seventeen neurodevelopmental outcomes were reported across five reviews [22,27,28,30–32]. One review reported that ACS exposure (unspecified number of courses) was associated with reduction in cerebral palsy in preterm born children compared to no ACS exposure [22]. Another review of observational studies reported that, compared to no ACS exposure, ACS was associated with reduction in childhood hearing impairment (OR 0.64, 95% CI 0.48 to 0.87, 7 studies, 8130 children) [31]. Metanalyses of other outcomes found no differences [22,27]. Two recent reviews evaluating effects of ACS in late preterm and term born children reported mixed results including an association with adverse neurodevelopmental outcomes as well as non-significant associations following ACS exposure [30,32].

*Psychological outcomes.* Five psychological outcomes were reported in five reviews [27,28,22,30]. Findings from observational studies suggested that these outcomes might vary with gestational age at birth [22,32]. The review reported that exposure to an unspecified number of ACS courses vs no ACS exposure for preterm birth might increase the risk of any mental or behavioural disorder in children with full-term birth [22]. However, no beneficial or adverse effect on psychological outcomes was observed in children with preterm birth. Another review also reported an increased prevalence of psychiatric problems among the ACS-exposed late-preterm and term born children compared to ACS-unexposed children [30].

*Physical growth outcomes.* Five growth outcomes were reported across three reviews [27,30,32]. In a growth-restricted population, ACS exposure compared to no ACS exposure might exhibit physical growth <10th percentile in early childhood (OR 5.20, 95% CI 1.40 to 19.62, 1 study, 91 children, low certainty evidence) [27]. Another review reported ACS was associated with a decrease in head circumference in children at age 6–10 years, compared to unexposed group (1 study, 116 children) [30]. A third review reported an increased odds of lower weight percentile in children at age ≥5 years who are born at term following ACS exposure [32].

*Respiratory and cardiovascular outcomes.* Three cardiovascular outcomes were reported in one review, though no meta-analyses were performed, nor were any differences reported [29]. Six respiratory outcomes, such as – asthma, wheeze, bronchiolitis were reported in one review for children who were born at term and found no differences between the groups [32].

*Survival/mortality outcomes.* One outcome (survival at school age) was reported in one review and no difference was observed between the groups [27].

**Repeat/multiple courses of ACS versus a single course of ACS.** *Neurodevelopmental outcomes.* Twenty-six neurodevelopmental outcomes were reported across five reviews [17,33–36]. Metanalyses found no differences between the repeat courses of ACS compared to a single course of ACS in the reported outcomes, namely neurodevelopmental/ neurocognitive impairment, developmental delay, cerebral palsy, blindness/visual impairment, and deafness/hearing impairment [17,34]. Another review found that, compared to a single course of ACS, at least one additional course of ACS might have an adverse effect on neurosensory outcomes in children born at term, while there were no long-term benefits or harms observed in preterm-born children [33].

*Psychological outcomes.* Eight psychological outcomes were reported in two reviews. Metanalyses showed no differences between the groups in psychological outcomes such as, abnormal child behaviour, mean scores of child behaviour assessment scales [17,34].

*Physical growth outcomes.* Sixteen child growth outcomes were reported in five reviews [17,33–36]. No differences were found for most of the growth outcomes, except in one 2022 review where a small reduction in mean weight during early

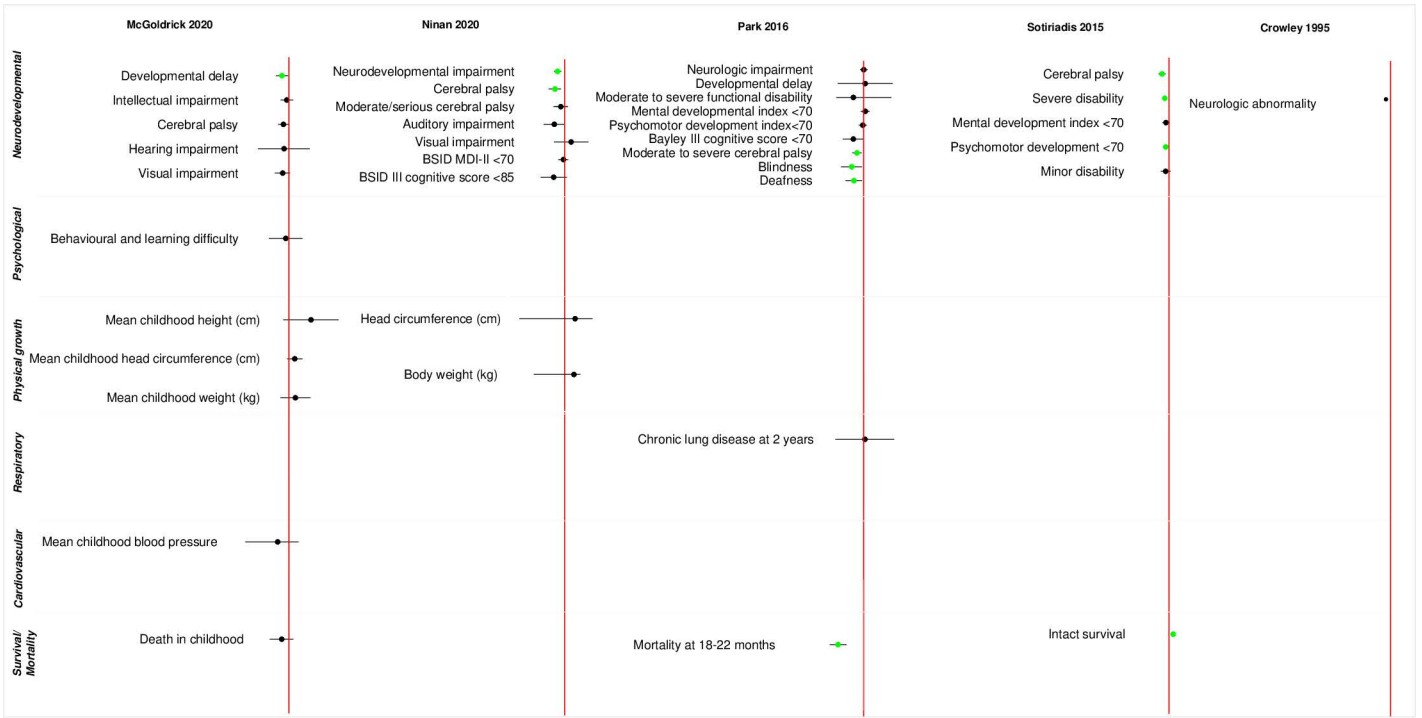

**Fig 2. Descriptive summary of meta-analysed reported outcomes (single course of ACS vs. placebo/no treatment).**

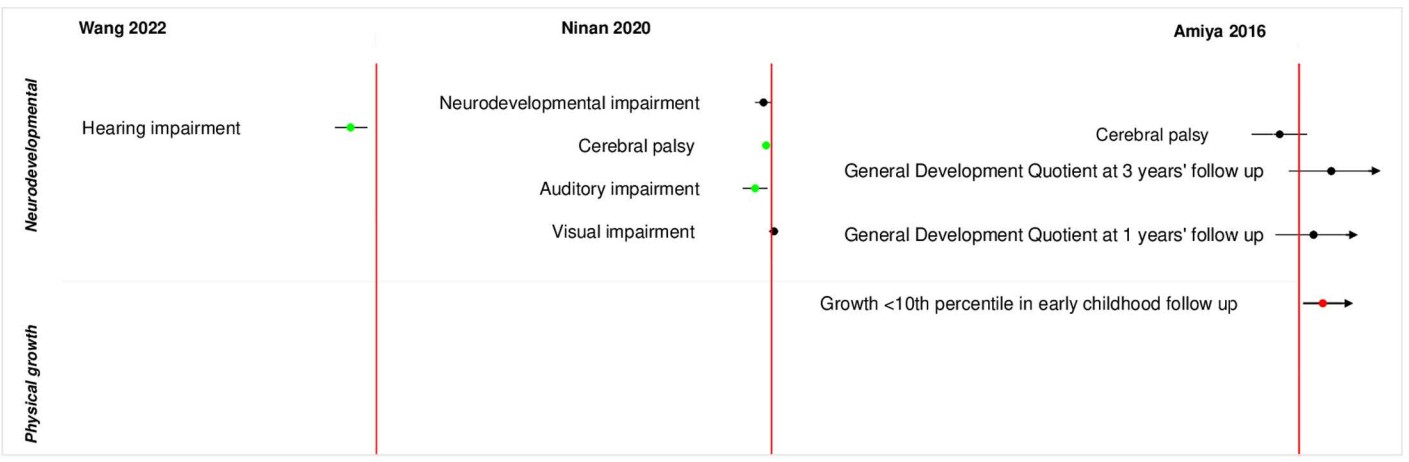

**Fig 3. Descriptive summary of meta-analysed reported outcomes (any ACS exposure vs. placebo/no ACS).**

childhood (2 to <5 years) in the repeat ACS group was observed (MD –0.16 kg, 95% CI –0.25 to –0.07, 4 studies, 3784 children, high-certainty evidence) [17]. Another review also reported a small reduction in weight z-scores during early childhood in the repeat ACS group (MD –0.11, 95% –0.19 to –0.03, 4 studies, 1878 children) [34].

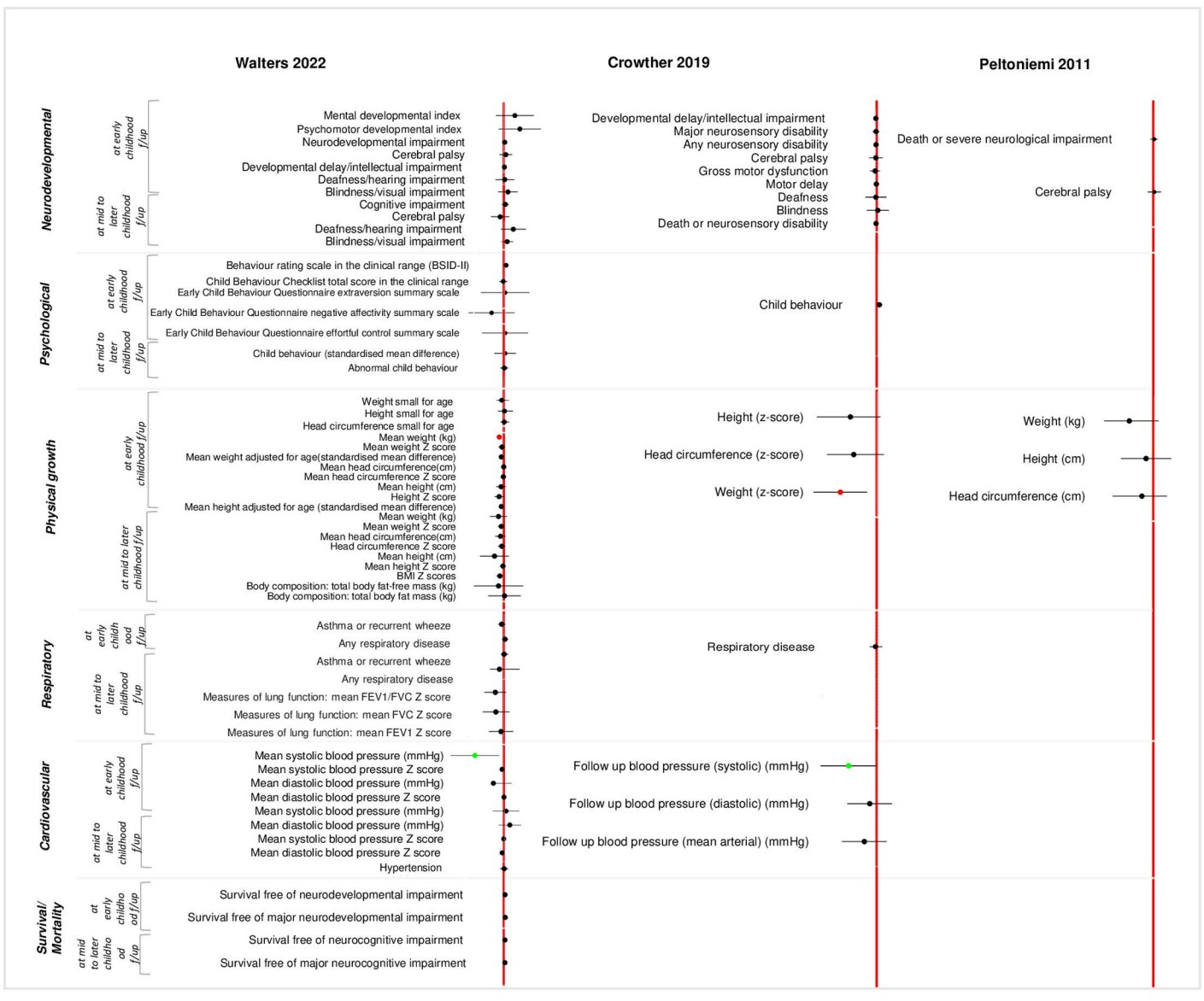

Green = positive health impact, Red = negative health impact, Black = no difference/mixed effects findings

**Fig 4. Descriptive summary of meta-analysed reported outcomes (repeat/multiple courses of ACS vs. single course of ACS).**

*Respiratory and cardiovascular outcomes.* Five respiratory and six cardiovascular outcomes were reported across two reviews [17,34]. Metanalyses of the reported respiratory outcomes (such as, asthma or recurrent wheeze, any respiratory diseases, measures of lung functions – FEV1, FVC Z, etc.) found no differences between the groups. However, a small reduction in mean systolic blood pressure was reported in the repeat ACS group in children during early childhood follow up at age <5 years [17,34].

*Survival/mortality outcomes.* Mortality outcome in children was reported across two reviews and they found no differences between the groups for mortality during the time of follow up at ages 2 years to 8 years [17,34].

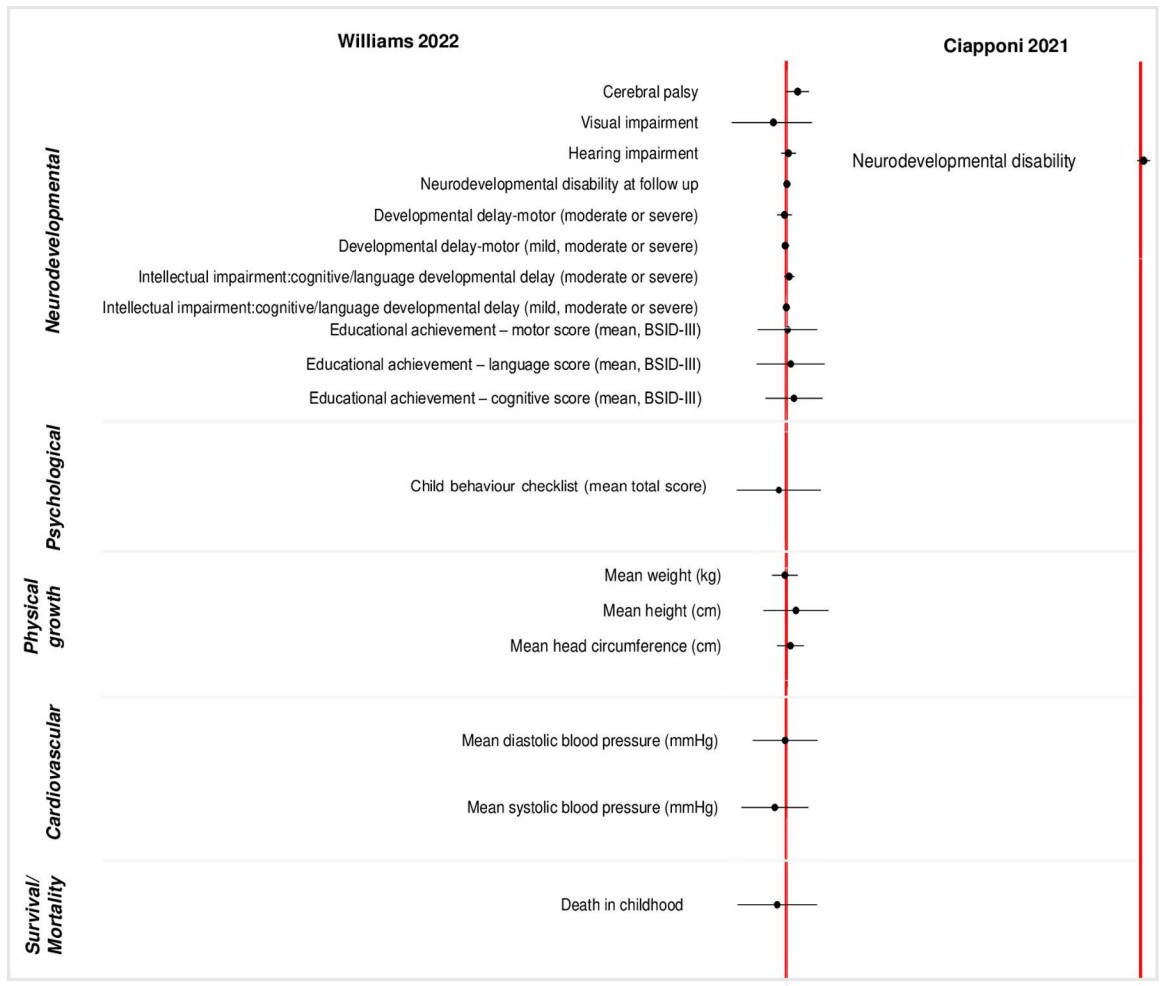

Green = positive health impact, Red = negative health impact, Black = no difference/mixed effects findings

**Fig 5. Descriptive summary of meta-analysed reported outcomes (ACS vs ACS).**

**ACS versus ACS.** *Neurodevelopmental outcomes.* Twelve neurodevelopmental outcomes were reported across two reviews [37,38]. Metanalyses found no clear differences between the groups (i.e., dexamethasone vs. betamethasone) in the reported outcomes, such as – neurodevelopmental disability, motor and cognitive developmental delay, cerebral palsy, hearing and vision impairment.

*Psychological outcomes.* One outcome (mean childhood behaviour checklist score) was reported in one review, though no differences were reported [37].

*Physical growth outcomes.* Three growth-related outcomes were reported in one review – no differences were observed in the reported outcomes (height, weight and head circumference at 2 years) between the groups [37].

*Respiratory and cardiovascular outcomes.* Two cardiovascular outcomes were reported in one review and no differences were observed [37]. No respiratory outcome was reported in the included reviews.

*Survival/mortality outcomes.* Only one outcome (childhood death at 2 years) was reported in a review and no difference was found between the groups [37].

## Discussion

In this overview of systematic reviews, we identified 19 systematic reviews that reported on child health outcomes following ACS administration for preterm birth. Within these reviews, a total of 75 unique primary studies were identified (13 trial follow-up studies and 62 observational studies). The majority of these 75 studies were conducted in high-income countries, indicative of a lack of evidence on child health outcomes from low- and middle-income countries where the majority of preterm births occur. The included reviews reported on a total of 106 unique child health outcomes. The outcomes varied widely in terms of operational definitions, terminology, timing of assessment, and measurement tools. These crossed six domains – neurodevelopmental, psychological, physical growth, respiratory, cardiovascular, and survival/mortality outcomes.

There is a generalized lack of robust evidence on the long-term effects of ACS. Although the 19 reviews included in this overview reported a varied range of effects of ACS on child health outcomes, for majority of the outcomes data were derived from only one or two studies, up to a maximum of seven studies; all outcomes had modest small sample sizes, which might be underpowered to detect clinically meaningful differences. Importantly, the majority of primary studies were observational designs, which are prone to multiple forms of bias and are generally considered low-quality evidence in clinical guideline development [39]. In this context, particularly when exposure and outcome data are collected retrospectively, the varying exposure to postnatal interventions that might impact important neurodevelopmental outcomes, including postnatal corticosteroids for reducing bronchopulmonary dysplasia [40], is not accounted for. Further, though the included reviews are relatively recent, the studies they include date back to as early as 1972, a time when neonatal intensive care was less advanced and robust as it is today. Consequently, findings and long-term morbidity outcomes from that period may differ from those observed in current settings. Additionally, considering outcomes only for a selected subgroup (e.g., a gestational age subgroup, such as preterm infants only) may introduce selection bias, potentially masking true treatment effects. When comparing different subgroups, it is essential to exercise caution by analyzing both treatment and control groups, rather than examining subgroups within only one group. Thus, the body of available evidence is unable to conclusively determine the long-term effects of ACS in children, across all outcome categories.

Available - mostly observational - evidence is that a single course of ACS might reduce developmental delay, cerebral palsy, neurodevelopmental disabilities, and intact survival in children (moderate to low certainty evidence). Whereas, repeat or multiple courses of ACS (compared to a single course of ACS) do not appear to worsen neurodevelopmental, psychological, respiratory, cardiovascular or mortality outcomes in children, though it causes reduced weight in early childhood (high certainty evidence). A Cochrane overview of reviews examining intrapartum interventions for preventing cerebral palsy also reported that compared to a single course of ACS, repeat doses of ACS possibly do not have an effect on the risk of cerebral palsy in children [41]. Low-certainty evidence suggests that there might have adverse effects on psychological, physical growth and some neurodevelopmental outcomes in children, particularly who are exposed to ACS prior to 34 weeks' gestation and subsequently born in the late preterm period or at term. The seemingly conflicting effects of ACS in children likely reflect the complex relationship between the timing of ACS administration and the gestational age at birth. While the therapeutic benefits of ACS are more critical for infants born at early preterm by reducing morbidity and mortality, these benefits may not extend to infants born at late preterm or term. One possible explanation is that as foetuses near term, they are naturally exposed to increasing levels of endogenous maternal and fetal cortisol. The added prolonged exposure to high doses of exogenous ACS may disrupt normal hormonal regulation, and potentially negatively impacting brain development and function [42]. Studies in animal models support this concern, showing that high/repeated doses of corticosteroids can negatively affect brain growth, body size and neurocognitive development [43]. Based on current knowledge, there are no clear differences between betamethasone or dexamethasone in terms of long-term child health outcomes. However, subgroup analysis in a separate review suggests that compared to dexamethasone, betamethasone might have a stronger protective effect on hearing impairment in extremely preterm born children [31].

Administration of ACS is now a standard of care for women at risk of preterm birth before 34 weeks' gestation and considered as an effective intervention for improving neonatal survival and prevention of respiratory distress syndrome

[44,45]. Nevertheless, there is growing concern on the longer-term health risks of ACS – exposed children born after 34 weeks', whether late preterm or at term, particularly for their neurodevelopmental and behavioural outcomes [22,30,32,33]. A recent population-based cohort study in Taiwan further suggests that compared to preterm-born infants, term-born infants have a greater risk of serious infections in the first 12 months of life following exposure to a single course of ACS [46]. The challenge of accurately predicting preterm birth often leads to over-treatment, exposing many infants to potential adverse effects of ACS without the intended benefits. Notably, approximately 30–40% of infants who receive ACS before 34 weeks' gestation are ultimately born at late preterm or term [32]. This inherent uncertainty under-scores the need for a more cautious and selective approach to ACS administration to minimize unnecessary exposure. Therefore, there is a critical need for further high-quality research to evaluate long term effects following ACS exposure, particularly in children born late preterm or at term, including follow up studies in low resource settings. Meanwhile, clini-cians and health professionals should be vigilant on the likelihood of birth after 34 weeks' before administering ACS, to optimize outcomes and reduce potential harms.

An overarching finding is the need for consistent outcome measurements in child follow up studies, across all domains. Previously, the Global Obstetric Network (GONet) has published 13 core outcome sets related to mother and offspring for preterm birth intervention trials. These include mortality, infection, early and late neurodevelopmental morbidity, respiratory morbidity and possible harms to offspring [47]. However, they did not specify operational definitions for these outcomes, nor how and when they should be measured. Precise outcome definitions, specification of time intervals for outcome evaluation, and standardized assessment methods will enable consistent outcome reporting in ACS follow-up studies. This will facilitate comparability between and across ACS interventions, and enhance understanding on longer-term effects of ACS in children.

Understanding the long-term effects of ACS during adulthood is also critical, given the growing recognition of early-life events on the development of chronic diseases and the potential role of glucocorticoid exposure in these relationships [48]. Preclinical animal studies have demonstrated that ACS exposure is associated with increased risks of cardiovascular, meta-bolic and other diseases in adult offspring [49–52]. However, human studies examining long-term cardiometabolic outcomes in adults exposed to ACS remain limited, and their findings are inconclusive, with most evidence derived from observational studies. Some studies found no association between ACS exposure and cardiometabolic disease risk in later life [53,54], while others reported elevated blood pressure, increased aortic stiffness, and dysregulated glucose metabolism [55–57]. Hence, longitudinal studies of trial cohorts are needed to definitively evaluate the effects of ACS throughout the life-course.

Our review has several strengths. We used a comprehensive search across multiple databases without restrictions in date of publication and language. We included systematic reviews including both RCTs and observational studies and included multiple types of comparisons – this allowed us to map a broad range of child health outcomes. Our review has some limitations. We did not undertake pooled analyses for the child health outcomes, considering the variability in the study designs, outcome measurement and reporting of effect estimates across the reviews and included primary studies. Another limitation is the quality of the included reviews that were rated as critically low to low confidence on AMSTAR. Additionally, the substantial overlap of primary studies in the included reviews might introduce bias and potentially impact the results. These limitations warrant caution in interpreting the findings.

## Conclusion

This overview summarizes evidence from available systematic reviews on long term child health outcomes following ACS administration for preterm birth. Based on current knowledge, ACS has no known deleterious effects on many child health outcomes, though further evidence is required. However, there is an emerging concern regarding potential adverse effects of ACS administration following late preterm and term birth. Variability in study designs and reported outcomes, along with few studies for many outcomes, underscore the uncertainty about the effects of ACS on many long-term outcomes. There is a need for high-quality follow up studies to generate a more robust evidence base on long term outcomes following ACS exposure.

## Supporting information

**S1 Table. AMSTAR-2 ratings of the included reviews.**
(DOCX)

**S2 Table. Summary of reported outcomes in the included reviews.**
(DOCX)

**S3 Table. Results of meta-analyses conducted by included systematic reviews.**
(DOCX)

**S1 Appendix. Search Strategy.**
(DOCX)

**S2 Appendix. Data extraction file.**
(XLSX)

**S3 Appendix. List of excluded studies with reason(s) for exclusion.**
(DOCX)

**S4 Appendix. Citation matrix.**
(DOCX)

## Author contributions

**Conceptualization:** Saima Sultana, Caroline SE Homer, Joshua P. Vogel.

**Data curation:** Saima Sultana, Fiona Bruinsma, Zeshi Fisher.

**Formal analysis:** Saima Sultana, Joshua P. Vogel.

**Investigation:** Fiona Bruinsma, Zeshi Fisher, Caroline SE Homer, Joshua P. Vogel.

**Methodology:** Saima Sultana, Caroline SE Homer, Joshua P. Vogel.

**Project administration:** Saima Sultana.

**Supervision:** Caroline SE Homer, Joshua P. Vogel.

**Validation:** Fiona Bruinsma, Zeshi Fisher, Caroline SE Homer, Joshua P. Vogel.

**Visualization:** Saima Sultana.

**Writing – original draft:** Saima Sultana.

**Writing – review & editing:** Fiona Bruinsma, Zeshi Fisher, Caroline SE Homer, Joshua P. Vogel.

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
