## [Decision Letter · Decision Letter 0]

8 Jan 2025

PGPH-D-24-02691

Long-term outcomes of antenatal corticosteroids for preterm birth: an overview of systematic reviews

Dear Dr. Sultana,

Thank you for submitting your manuscript to PLOS Global Public Health. After careful consideration, we feel that it has merit but does not fully meet PLOS Global Public Health’s publication criteria as it currently stands. Therefore, we invite you to submit a revised version of the manuscript that addresses the points raised during the review process.

The manuscript has been assessed by two reviewers and their comments are available below. The reviewers are generally positive about the manuscript, but they do have a few suggestions and requests for clarification for your consideration. Could you please carefully revise the manuscript to address the concerns raised?

We look forward to receiving your revised manuscript.

Kind regards,

Marianne Clemence

Staff Editor

Journal Requirements:

1. As required by our policy on Data Availability, please ensure your manuscript or supplementary information includes the following: 

Additional Editor Comments (if provided):

Reviewers' comments:

Reviewer's Responses to Questions

**Comments to the Author**

1. Does this manuscript meet PLOS Global Public Health’s publication criteria?

Reviewer #1: Yes

Reviewer #2: Yes

2. Has the statistical analysis been performed appropriately and rigorously?

Reviewer #1: Yes

Reviewer #2: Yes

3. Have the authors made all data underlying the findings in their manuscript fully available (please refer to the Data Availability Statement at the start of the manuscript PDF file)?

Reviewer #1: Yes

Reviewer #2: Yes

4. Is the manuscript presented in an intelligible fashion and written in standard English?

Reviewer #1: Yes

Reviewer #2: Yes

Reviewer #1: Dear Authors

Your review is very interesting But I have some recommends

1. In Your review have some contradictory data according to multiple course of ACST and its relation to neurodevelopmental in neonate, childhood as well as in the adults.

2. Does Authors recommend single or multiple courses of ACST in preventing long time sequels.

3. Authors did not emphases long-time sequels and differences between single maternal beta and dexamethasone therapy

relation to cardiovascular and metabolism in adults.

Reviewer #2: Very rigorously done and well written! Congratulations.

Table 1 :

-Sotiriadis : « the effects », instead of « to effects », or “to assess effects” ; sometimes it is written “to assess the effects” and sometimes “to assess effects”, could you make it uniform?

Outcomes in the systematic reviews are often assessed within a subgroup of the population that received antenatal corticosteroids, be it only in the premature born children or in the (late pre)term born children. Looking at outcomes of children within such subgroups, poses a major risk of selection bias. This needs to be mentioned in the discussion.

**Do you want your identity to be public for this peer review?** For information about this choice, including consent withdrawal, please see our Privacy Policy

Reviewer #1: No

Reviewer #2: **Yes: ** Isabelle Dehaene

---

## [Decision Letter · Decision Letter 1]

10 Mar 2025

PGPH-D-24-02691R1

Long-term outcomes of antenatal corticosteroids for preterm birth: an overview of systematic reviews

Dear Dr. Sultana,

Thank you for submitting your manuscript to PLOS Global Public Health. After careful consideration, we feel that it has merit but does not fully meet PLOS Global Public Health’s publication criteria as it currently stands. Therefore, we invite you to submit a revised version of the manuscript that addresses the points raised during the review process.

The manuscript has been re-evaluated by one of the two original reviewers, who requests one minor revision.

We look forward to receiving your revised manuscript.

Kind regards,

Steve Zimmerman, PhD

PLOS Staff Editor

Journal Requirements:

Additional Editor Comments (if provided):

Reviewers' comments:

Reviewer's Responses to Questions

**Comments to the Author**

Reviewer #2: (No Response)

publication criteria?

Reviewer #2: Yes

3. Has the statistical analysis been performed appropriately and rigorously?

Reviewer #2: Yes

4. Have the authors made all data underlying the findings in their manuscript fully available (please refer to the Data Availability Statement at the start of the manuscript PDF file)?

Reviewer #2: No

5. Is the manuscript presented in an intelligible fashion and written in standard English?

Reviewer #2: Yes

Reviewer #2: Could you please change the sentence you added by this sentence (or an alternative of your choice which covers the topic)?

Additionally, assessing outcomes only in specific subgroups (e.g., gestational age) within one treatment arm may introduce bias. Within these subgroups, the comparison is not between no treatment or placebo and treatment, but between the subgroups (all under treatment or no treatment/placebo), which cannot be subject of randomization.

**Do you want your identity to be public for this peer review?** For information about this choice, including consent withdrawal, please see our Privacy Policy

Reviewer #2: **Yes: ** Isabelle Dehaene

---

## [Editor Report · Decision Letter 2]

9 Apr 2025

Long-term outcomes of antenatal corticosteroids for preterm birth: an overview of systematic reviews

PGPH-D-24-02691R2

Dear Dr Sultana,

We are pleased to inform you that your manuscript 'Long-term outcomes of antenatal corticosteroids for preterm birth: an overview of systematic reviews' has been provisionally accepted for publication in PLOS Global Public Health.

Best regards,

Julia Robinson

Executive Editor